# Generated and Pseudo Content guided Prototype Refinement for Few-shot Point Cloud Segmentation

**Lili Wei**[1,2]   **Congyan Lang**[1,2*]   **Ziyi Chen**[1,2]   **Tao Wang**[1,2]   **Yidong Li**[1,2]   **Jun Liu**[3]

[1] School of Computer Science & Technology, Beijing Jiaotong University, China
[2]Key Laboratory of Big Data & Artificial Intelligence in Transportation, Ministry of Education, China
[3]School of Computing and Communications, Lancaster University, UK
{20112014, cylang, 20112021, twang, ydli}@bjtu.edu.cn
j.liu81@lancaster.ac.uk

## Abstract

Few-shot 3D point cloud semantic segmentation aims to segment query point clouds with only a few annotated support point clouds. Existing prototype-based methods learn prototypes from the 3D support set to guide the segmentation of query point clouds. However, they encounter the challenge of low prototype quality due to constrained semantic information in the 3D support set and class information bias between support and query sets. To address these issues, in this paper, we propose a novel framework called **G**enerated and **P**seudo **C**ontent guided **P**rototype **R**efinement (GPCPR), which explicitly leverages LLM-generated content and reliable query context to enhance prototype quality. GPCPR achieves prototype refinement through two core components: LLM-driven Generated Content-guided Prototype Refinement (GCPR) and Pseudo Query Context-guided Prototype Refinement (PCPR). Specifically, GCPR integrates diverse and differentiated class descriptions generated by large language models to enrich prototypes with comprehensive semantic knowledge. PCPR further aggregates reliable class-specific pseudo-query context to mitigate class information bias and generate more suitable query-specific prototypes. Furthermore, we introduce a dual-distillation regularization term, enabling knowledge transfer between early-stage entities (prototypes or pseudo predictions) and their deeper counterparts to enhance refinement. Extensive experiments demonstrate the superiority of our method, surpassing the state-of-the-art methods by up to 12.10% and 13.75% mIoU on S3DIS and ScanNet, respectively.

## 1   Introduction

Point cloud semantic segmentation, aiming to assign a semantic label to each point in the 3D point clouds, benefits various applications like autonomous driving, robotics, and augmented/virtual reality. Despite the remarkable performance of supervised methods in this field [33, 34, 22, 24, 12, 37], they rely on labor-intensive labeled data and struggle to distinguish novel classes not present in the training data. To address these issues, few-shot 3D point cloud semantic segmentation (FS-3DSeg), which segments novel classes with only a few labeled data, has gained widespread attention.

FS-3DSeg aims to learn a model on base classes and generalize it to novel classes with only a few annotated point clouds. Existing FS-3DSeg methods [39, 17, 36, 9, 41, 20, 10, 16] typically adopt prototype-based paradigms [28, 32], where prototypes extracted from the support point clouds are treated as class descriptors to guide the segmentation of unlabeled query point clouds by matching

---

*Corresponding author.

38th Conference on Neural Information Processing Systems (NeurIPS 2024).

query point features with prototypes. Due to their representation and generalization capabilities, prototype-based methods have made great progress and garnered increasing attention. However, their performance remains unsatisfactory due to encountering the challenge of low prototype quality, attributed to two reasons: 1) **Semantic information constraints**: Limited 3D support point clouds contain only partial and incomplete object information, lacking both intra-class diversity and inter-class discriminative information. Consequently, vanilla prototypes generated from the 3D support set are insufficient to be comprehensive class descriptors. 2) **Class information bias**: Query and support point clouds exhibit intra-class object variations [16, 20] and feature distribution gaps [9, 10], making vanilla prototypes may be ill-suited for segmenting query samples as the distances between vanilla prototypes and query features will be far away.

To address these issues, we propose to compensate for the lack of semantics in 3D support set to generate comprehensive and reliable query-specific prototypes for accurately segmenting the query point cloud. This can be achieved in two steps. Firstly, to alleviate class information constraints, motivated by the fact that texts are more controllable and accessible [35], we aim to integrate general semantic knowledge from texts to refine prototypes. Fortunately, Large Language Models (**LLM**) (*e.g.*, ChatGPT) [3, 35, 29] offer promising opportunities to help achieve this goal due to their wealth of implicit knowledge and have been leveraged for a variety of downstream tasks, such as image segmentation [42] and point cloud classification [43]. Inspired by this, we leverage the capabilities of LLM to explicitly generate diverse and differentiated contents, combined with novel designs to obtain refined prototypes with richer semantics. Secondly, to reduce class information bias, previous works [9, 20, 16, 10] have endeavored to map prototypes to query feature space through prototype-query interaction using cross-attention [31]. However, noisy interactions between prototypes of one class and query features of other classes may introduce class-irrelevant cues, thereby undermining the reliability of prototypes. We observed that pseudo masks for query point clouds can assist in extracting class-specific pseudo-query context and filtering out class-irrelevant noisy features.

Ultimately, in this paper, we propose a novel FS-3DSeg framework via **G**enerated and **P**seudo **C**ontent guided **P**rototype **R**efinement (**GPCPR**), which progressively refines prototypes by two newly proposed components: LLM-driven Generated Content-guided Prototype Refinement (GCPR) and Pseudo Query Context-guided Prototype Refinement (PCPR). Initially, vanilla prototypes are generated from 3D support features. Firstly, our GCPR prompts the LLM to generate diverse descriptions of single classes and differentiated descriptions of class pairs. Then we design text adapters and two text-to-prototype compressors for mapping and incorporating text knowledge into vanilla prototypes, yielding refined prototype covering with richer and discriminative semantic knowledge. Secondly, leveraging the refined prototypes, PCPR generates pseudo masks for query point clouds to derive the class-specific pseudo-query context, which can be aggregated into prototypes by a newly designed query-to-prototype compressor, yielding more reliable prototypes that can better associate with query points. Beyond that, we also introduce a dual-distillation regularization term, which enables early-stage entities (prototypes or pseudo predictions) to gain insight from their deeper optimal counterparts. This facilitates bidirectional information exchange in both forward and backward processes during training, thus obtaining better prototypes and predictions.

We summarise our contributions as follows: 1) We propose GPCPR, a novel end-to-end FS-3DSeg framework that enhances prototype quality by simultaneously integrating LLM-generated content and reliable query context to generate query-specific prototypes. To the best of our knowledge, this is the first time leveraging LLM's capabilities to segment novel classes in FS-3DSeg. 2) We design a series of novel modules, including the Generated Content-guided Prototype Refinement (GCPR) module and the Pseudo Query Context-guided Prototype Refinement (PCPR) module, to facilitate the prototype refinement process. Additionally, we design a dual-distillation regularization term to further mutually enhance the refinement. 3) Extensive experiments demonstrate the superiority of our method, notably exceeding state-of-the-art methods by up to 12.10% and 13.75% on S3DIS and ScanNet datasets, respectively.

## 2 Related Work

### 2.1 3D Point Cloud Semantic Segmentation

3D point cloud semantic segmentation (3DSeg) aims to assign semantic labels to each point within point clouds. Supervised 3DSeg studies involve voxel-based methods [7, 18] and point-based methods

[22, 33, 23, 24, 12, 37], with the latter receiving increasing attention due to the simplicity, flexibility, and efficiency. For example, DGCNN [33] proposes the EdgeConv module to capture local structures. Recently, [12, 37] design self-attention-like networks to model long-range contexts from distant neighbors. Despite their effectiveness, these methods often rely on extensive annotations and cannot segment novel classes. In this paper, following mainstream FS-3DSeg methods [39, 9, 16], we utilize DGCNN as the point encoder and extend its capability to segment novel classes.

## 2.2 Few-shot 3D Point Cloud Segmentation

Most few-shot 3D point cloud semantic segmentation (FS-3DSeg) methods [39, 17, 9, 20, 10, 41, 16, 1] follow the prototype-based paradigms [32, 28, 11], learning prototypes from the support set to segment query point clouds. Specifically, AttMPTI [39], the pioneering FS-3DSeg method, proposes an attention-aware multi-prototype transductive inference framework based on label propagation. BFG [17] mutually embeds global perception into local features and prototypes. To reduce contextual gaps between support prototypes and query features, [20, 10, 9, 16] adapt prototypes to query feature space by prototype-query feature interaction via cross attention [31]. COSeg [1] introduces correlation optimization by refining the multi-prototypical support-query correlations. In addition to prototype learning, SCAT [36] explores class-specific relations between query and support features using transformer blocks [31] instead of pooling operations. However, these methods often yield low-quality prototypes since the 3D support set only contains constrained semantic information. Besides, their prototype-query interaction [20, 10, 9, 16] may introduce noise. In contrast, our approach refines prototypes by integrating LLM-generated content and class-specific pseudo-query context, resulting in well-suited query-specific prototypes for better segmentation.

## 2.3 Large Models

Recently, several large models have emerged, including large language models (*e.g.*, GPT-3 [3], ChatGPT [35] and BERT [5]) and large multi-modal models (*e.g.*, CLIP [25], BLIP [13], LLaVA [15], GPT-4 Vision [35] and MiniGPT [40]). These extensive models are repositories of extensive knowledge and have achieved widespread success in diverse downstream tasks, such as robot task plans [27], image classification [26] and image segmentation [42]. In the field of 3D point cloud understanding, several large models have also emerged that associate point clouds, texts and other modalities (*e.g.*, Point-LLM [8], MiniGPT-3D [29], Uni3D-LLM [14] and PointCLIP V2 [43]). However, none of them are designed for segmenting novel classes in FS-3DSeg. In this work, we leverage the rich generated content of LLM to assist in addressing the FS-3DSeg problem.

# 3 Method

## 3.1 Problem Formulation and Overview

**Problem Formulation.** Following previous FS-3DSeg methods [39, 9], we adopt meta-learning based paradigm. Specifically, all semantic classes are divided into base class set $C_{base}$ and novel class set $C_{novel}$ for training and testing respectively, where $C_{base} \cap C_{novel} = \emptyset$. Each few-shot task (*i.e.*, episode) instantiates an $N$-way $K$-shot segmentation task. The input data in each episode contains a support set $S = \{(\mathbf{I}_s^{n,k}, \mathbf{M}_s^{n,k})_{k=1}^{K}\}_{n=1}^{N}$ and a query set $Q = \{(\mathbf{I}_q^i, \mathbf{M}_q^i)\}_{i=1}^{T}$, where $N$, $K$ and $T$ denote the number of classes, the number of support point clouds for each class, and the number of query point clouds. $\mathbf{I}_s^{n,k}$ and $\mathbf{I}_q^i$ denote support and query point clouds, each contains $M$ points. $\mathbf{M}_s^{n,k} \in \{0,1\}^{M \times 1}$ denotes binary support GT mask for each of the $N$ unique classes. $\mathbf{M}_q^i \in \{0,...,N\}^{M \times 1}$ denotes the query GT mask, which is only available during training. The class name set contains one background name and $N$ foreground class names, denoted as $C = \{c_n\}_{n=0}^{N}$. The goal of FS-3DSeg is to learn a model to predict segmentation mask $\hat{\mathbf{M}}_q$ for $\mathbf{I}_q$ based on $S$.

**Overview.** Figure 1 illustrates the architecture of our proposed method. Given query set and support set $Q$ and $S$, following common practice [39, 9], we utilize a shared point encoder to extract per point query and support features, represented as $\mathbf{F}_q \in \mathbb{R}^{T \times M \times d}$ and $\mathbf{F}_s \in \mathbb{R}^{N \times K \times M \times d}$, where $d$ denotes feature dimension. Then we adopt masked average pooling (MAP) [9] to generate vanilla class-wise 3D support prototypes $\mathbb{P} = \{\mathcal{P}^n\}_{n=0}^{N} \in \mathbb{R}^{(N+1) \times d}$ from $\mathbf{F}_s$, including a background prototype and $N$ foreground prototypes. To enhance the quality of $\mathbb{P}$, we propose two modules: LLM-

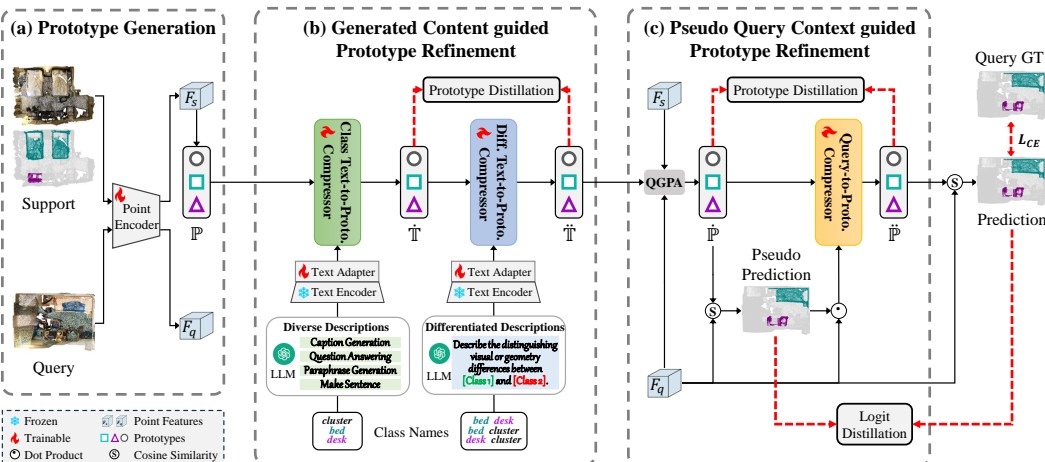

Figure 1: Overview of our method. The support flow is responsible for Prototype Generation, LLM-driven Generated Content-guided Prototype Refinement (GCPR), and Pseudo Query Context-guided Prototype Refinement (PCPR). Dual-distillation, *i.e.*, prototype distillation and logit distillation, further enhance the refinement process. The query flow measures distances between query point features and refined prototypes to predict segmentation results.

driven Generated Content-guided Prototype Refinement (GCPR) and Pseudo Query Context-guided Prototype Refinement (PCPR). Firstly, GCPR prompts LLM to generate diverse class descriptions $D$ and differentiated class descriptions $D^{'}$, followed by a frozen text encoder and a trainable text adapter to extract text features $\mathbf{E}$ and $\mathbf{E}^{'}$. To effectively incorporate text knowledge into $\mathbb{P}$, GCPR sequentially utilizes two text-to-prototype compressors to generate refined prototypes $\ddot{\mathbb{T}}$ with rich semantics. After that, PCPR generates pseudo masks $\dot{\mathbf{M}}_q$ to extract class-specific pseudo query contexts $\dot{\mathbf{F}}_q$ for $Q$, followed by a query-to-prototype compressor to generate well-suited query-specific prototypes $\ddot{\mathbb{P}}$. Finally, cosine similarities between $\mathbf{F}_q$ and $\ddot{\mathbb{P}}$ are calculated to segment query point clouds, where each point cloud is assigned the label of the most similar prototype. To supervise the model training, we adopt standard cross-entropy loss and design a dual-distillation regularization term (DD loss) to further enhance the refinement process. Subsequently, we provide detailed descriptions of GCPR, PCPR, and DD loss as below.

## 3.2 LLM-driven Generated Content-guided Prototype Refinement

Due to support point clouds only containing constrained semantic information, the vanilla prototypes $\mathbb{P}$ lack class diversity and cannot be better associated with $Q$. To address this, inspired by the common-sense knowledge in LLM and the fact that textual descriptions are more controllable [35], we aim to leverage LLM to generate detailed class content and propose two text-to-prototype compressors to embed these texts into prototypes $\mathbb{P}$, thereby obtaining refined prototypes with richer semantics.

**LLM-driven Content Generation.** By leveraging the extended world knowledge of LLM (*e.g.*, GPT-3 [3]), we generate two types of content at scale, *i.e.*, diverse class descriptions and differentiated class descriptions, respectively for enhancing the comprehensiveness and discriminability of the prototypes $\mathbb{P}$.

1) *Diverse Class Descriptions.* To generate diverse descriptions, effective prompt templates for LLMs are essential. Inspired by the heuristic 3D-oriented commands for depth maps in [43], we designed the following 3D-specific heuristic prompt templates to ensure that the texts encompass sufficient point cloud properties:

```
Caption Generation: "Describe a point cloud of a [CLASS] in one sentence."
Question Answering: "What is a [CLASS] point cloud like?"
Paraphrase Generation: "Generate a synonym: A point cloud of a [CLASS]."
Make Sentence: "Make a sentence with words: point cloud, [CLASS], obscure."
```

For each command, we place "[CLASS]" with a class name and feed it into GPT-3 to generate $N_{div}$ descriptions. The generated description set can be represented as: $D = \{D_{c_n}\}_{n=0}^{N}$, where $D_{c_n}$ contains $4 \times N_{div}$ descriptions.

2) *Differentiated Class Descriptions.* To enhance the model's ability to distinguish between multiple classes, inspired by [6] which generate differentiated attribute descriptions between ambiguous classes for image classification, we intend to generate 3D-specific differentiated descriptions for any pair of classes, explaining their visual and geometric differences. To this end, we designed the following template to prompt LLM to generate differentiated class descriptions:

```
"Describe the distinguishing visual or geometry differences between the
point clouds of [CLASS 1] and [CLASS 2] in pairs of sentences. Generate as
many captions as you can."
```

We place "[CLASS 1]" and "[CLASS 2]" with a pair of class names and trigger GPT-3 to produce several sentences, each of which is split into two parts at the conjunctions, with the first (or second) part assigned as a differentiated description for class $c_1$ (or $c_2$), denoted as $D_{c_1}^{c_2}{}'$ (or $D_{c_2}^{c_1}{}'$). For each class, we combine all its differentiated descriptions with other classes to contain more discriminative information, forming a differential description set $D' = \{D'_{c_n}\}_{n=0}^{N}$, where $D'_{c_n} = \{D_{c_n}^{c_i}{}'\}_{c_i \in C \setminus \{c_n\}}$ includes $N_{diff}$ descriptions.

Detailed explanations and examples for diverse/differentiated descriptions are given in Appendix A.

**Text-To-Prototype Compressors.** Given LLM-generated content $D$ and $D'$, we employ a frozen CLIP textual encoder [25] and a shared text adapter to extract text features $\mathbf{E} \in \mathbb{R}^{(N+1) \times (4 \times N_{div}) \times d}$ and $\mathbf{E}' \in \mathbb{R}^{(N+1) \times N_{diff} \times d}$. To incorporate text knowledge into vanilla prototypes $\mathbb{P}$, we propose two cross-attention-based [31] text-to-prototype compressors, *i.e.*, the class text-to-prototype compressor and the differentiated text-to-prototype compressor, which measure the relations between $\mathbb{P}$ and texts to aggregate semantic information.

*1) Class text-to-prototype compressor* integrates features $\mathbf{E}$ for diverse descriptions into vanilla prototypes $\mathbb{P}$ to obtain refined prototypes $\dot{\mathbb{T}} = \{\dot{\mathcal{T}}^n\}_{n=0}^{N}$ with richer semantics, each formulated as:

$$\dot{\mathcal{T}}^n = \mathcal{P}^n + softmax(\mathbf{Q_1}\mathbf{K_1}^\top)\mathbf{V_1}, \quad n \in \{0, ..., N\}, \tag{1}$$

where $\mathbf{Q_1} = \mathcal{P}^n W_{q1} \in \mathbb{R}^{1 \times d'}$, $\mathbf{K_1} = \mathbf{E}^n W_{k1} \in \mathbb{R}^{(4 \times N_{div}) \times d'}$ and $\mathbf{V_1} = \mathbf{E}^n W_{v1} \in \mathbb{R}^{(4 \times N_{div}) \times d}$. $W_{q1}, W_{k1} \in \mathbb{R}^{d \times d'}$ and $W_{v1} \in \mathbb{R}^{d \times d}$ are learnable parameters for fully connected layers. $d'$ denotes projected feature dimension.

*2) Differentiated text-to-prototype compressor* integrates features $\mathbf{E}'$ for differentiated descriptions into previous prototypes $\dot{\mathbb{T}}$ to obtain refined prototypes $\ddot{\mathbb{T}} = \{\ddot{\mathcal{T}}^n\}_{n=0}^{N}$ with discriminative information, each formulated as:

$$\ddot{\mathcal{T}}^n = \dot{\mathcal{T}}^n + softmax(\mathbf{Q_2}\mathbf{K_2}^\top)\mathbf{V_2}, \quad n \in \{0, ..., N\}, \tag{2}$$

where $\mathbf{Q_2} = \dot{\mathcal{T}}^n W_{q2} \in \mathbb{R}^{1 \times d'}$, $\mathbf{K_2} = \mathbf{E}^{n'} W_{k2} \in \mathbb{R}^{N_{diff} \times d'}$, and $\mathbf{V_2} = \mathbf{E}^{n'} W_{v2} \in \mathbb{R}^{N_{diff} \times d}$. $W_{q2}, W_{k2} \in \mathbb{R}^{d \times d'}$ and $W_{v2} \in \mathbb{R}^{d \times d}$ are learnable parameters for fully connected layers.

### 3.3 Pseudo Query Context-guided Prototype Refinement

Query and support point clouds often exhibit large object variations and class information bias, making vanilla support prototypes $\mathbb{P}$ unsuitable for segmenting query point clouds. Although our GCPR compensates for the lack of semantics $\mathbb{P}$, they still lack query-specific associations. Previous approaches have endeavored to customize query-specific prototypes through prototype-query feature interaction using cross-attention [31]. Specifically, QGPNet [10] and QGPA [9] align support prototypes with query features to reduce the channel-wise feature distribution gap but lack query-specific context. QGE [20] and DPA [16] aggregate query context into support prototypes, but inevitably introduce noisy interactions between prototypes of one class and query features of other classes. This could inadvertently incorporate class-irrelevant query cues into prototypes, thereby undermining their reliability and exacerbating the challenge of distinguishing confusing classes. To address this issue, we aim to generate more reliable class-specific pseudo query context from query features and propose a transformer-based query-to-prototype compressor to customize well-suited query-specific prototypes.

**Pseudo-Query Context Generation.** To extract class-specific context, we need to predict pseudo masks $\mathbf{M}_q^i$ for each query point cloud $\mathbf{I}_q^i$ and extract class-specific context $\dot{\mathbf{F}}_q^i$ accordingly, effectively filtering out interfering

cues from other classes. However, the channel distribution gap between prototypes $\ddot{\mathbb{T}}$ and query features $\mathbf{F}_q^i$ may lead to inaccurate pseudo masks. Thus we follow QGPA [9] to rectify current prototypes $\ddot{\mathbb{T}}$ to query feature channel distribution through channel-wise cross-attention [31], formulated as:

$$\dot{\mathcal{P}}^{i,n} = \ddot{\mathcal{T}}^n + softmax(\mathbf{Q_3}\mathbf{K_3}^\top)\mathbf{V_3}, \quad n \in \{0, ..., N\}, i \in \{1, ..., T\}, \tag{3}$$

where $\mathbf{Q_3} = \mathbf{F}_q^{i\top} W_{q3} \in \mathbb{R}^{d \times M'}$, $\mathbf{K_3} = \mathbf{F}_s^{n\top} W_{k3} \in \mathbb{R}^{d \times M'}$, and $\mathbf{V_3} = \ddot{\mathcal{T}}^n W_{v3} \in \mathbb{R}^{1 \times d}$. $\mathbf{F}_s^n \in \mathbb{R}^{M \times d}$ denotes averaged support features, calculated by $\mathbf{F}_s^n = 1/(NK) \cdot \sum_{N,k} \mathbf{F}_s^{n,k}$ if $n = 0$, otherwise $\mathbf{F}_s^n = 1/K \cdot \sum_k \mathbf{F}_s^{n,k}$. $W_{q3}, W_{k3} \in \mathbb{R}^{M \times M'}$ and $W_{v3} \in \mathbb{R}^{d \times d}$ are the learnable parameters for FC. Then query-specific prototypes $\dot{\mathbb{P}} = \{\dot{\mathbb{P}}^i\}_{i=1}^T \in \mathbb{R}^{T \times (N+1) \times d}$ are obtained, where $\mathbb{P}^i$ denote query-specific prototypes for each query point cloud $\mathbf{I}_q^i$.

Then, we calculate cosine similarity between $\mathbf{F}_q^i$ and prototypes $\dot{\mathbb{P}}^i$ to predict pseudo logits $\dot{\mathbf{L}}_q^i \in \mathbb{R}^{M \times (N+1)}$ and pseudo one hot masks $\dot{\mathbf{M}}_q^i \in \mathbb{R}^{M \times (N+1)}$. By masking query features with pseudo masks, we obtain class-specific pseudo query context, represented as: $\dot{\mathbf{F}}_q = \{\dot{\mathbf{F}}_q^n\}_{n=0}^N, \dot{\mathbf{F}}_q^n \in \mathbb{R}^{(T \times M) \times d}$.

**Query-To-Prototype Compressor.** The query-to-prototype compressor aggregates pseudo query context features into the prototypes by cross attention, represented as:

$$\ddot{\mathcal{P}}^{i,n} = \dot{\mathcal{P}}^{i,n} + softmax(\mathbf{Q_4}\mathbf{K_4}^\top)\mathbf{V_4}, \qquad n \in \{1, ..., N\}, \quad i \in \{1, ..., T\}, \tag{4}$$

where $\mathbf{Q_4} = \dot{\mathcal{P}}^{i,n} W_{q4} \in \mathbb{R}^{1 \times d'}$, $\mathbf{K_4} = \dot{\mathbf{F}}_q^n W_{k4} \in \mathbb{R}^{(T \times M) \times d'}$, $\mathbf{V_4} = \dot{\mathbf{F}}_q^n W_{v4} \in \mathbb{R}^{(T \times M) \times d}$. $W_{q4}, W_{k4} \in \mathbb{R}^{d \times d'}$ and $W_{v4} \in \mathbb{R}^{d \times d}$ are learnable parameters for fully connected layers. After that, we obtain the final query-specific prototypes well-suited to segment query point clouds, represented as $\ddot{\mathbb{P}} = \{\ddot{\mathbb{P}}^i\}_{i=1}^T \in \mathbb{R}^{T \times (N+1) \times d}$.

Finally, we calculate cosine similarity scores between $\mathbf{F}_q^i$ and $\ddot{\mathbb{P}}^i$ to obtain predicted logits $\hat{\mathbf{L}}_q^i \in \mathbb{R}^{M \times (N+1)}$. Each point is then assigned the label of the most similar prototype to generate the final predicted mask $\hat{\mathbf{M}}_q^i$.

## 3.4 Dual-Distillation Regularization

To further enhance prototype refinement, we introduce a dual-distillation regularization term that allows early-stage entities to gain insights from their deeper counterparts, including two types of self-distillation: *i.e.*, prototype distillation and pseudo prediction distillation.

**Prototype Distillation**. In the forward process, GCPR and PCPR integrate useful semantics into early-stage prototypes to generate refined prototypes. To achieve mutually beneficial and bi-directional optimization of multi-stage prototypes, we treat early-stage prototypes (*e.g.*, $\dot{\mathbb{T}}$ or $\dot{\mathbb{P}}$) as student prototypes and deep-stage prototypes (*e.g.*, $\ddot{\mathbb{T}}$ or $\ddot{\mathbb{P}}$) as teacher prototypes, employing prototype distillation in GCPR and PCPR, defined as:

$$\mathcal{L}_{TP} = KL(\dot{\mathbb{T}}||\ddot{\mathbb{T}}), \quad \mathcal{L}_{QP} = KL(\dot{\mathbb{P}}||\ddot{\mathbb{P}}), \tag{5}$$

where KL refers to the Kullback-Leibler divergence. The self-distillation process facilitates effective knowledge transfer from teacher prototypes to student prototypes, enhancing the refinement process.

**Pseudo Prediction Distillation**. In PCPR, pseudo masks play a core role as they determine the reliability of the generated class-specific pseudo query context. To further improve the accuracy and quality of pseudo masks, we treat pseudo logits $\dot{\mathbf{L}}_q$ as students and final predicted logits $\hat{\mathbf{L}}_q$ as teachers, and employ logit distillation in PCPR, formulated as:

$$\mathcal{L}_{QM} = KL(\dot{\mathbf{L}}_q||\hat{\mathbf{L}}_q). \tag{6}$$

## 3.5 Objective.

During training, the proposed model is supervised by two loss functions, *i.e.*, a standard cross-entropy loss $\mathcal{L}_{SEG}$ and the proposed dual-distillation loss (DD loss). The cross-entropy loss serves as the main optimization target, aimed at learning comprehensive and reliable query-specific prototypes. The dual-distillation loss acts as a regularization term to promote knowledge transfer between early-stage entities (prototypes or pseudo-predictions) and their deeper counterparts. Formally, the cross entropy loss is defined as:

$$\mathcal{L}_{SEG} = \mathcal{L}_{CE}(\dot{\mathbf{M}}_q, \hat{\mathbf{M}}_q). \tag{7}$$

The overall loss is a weighted combination of $\mathcal{L}_{SEG}$ and DD loss with a balancing weight $\lambda$, represented as:

$$\mathcal{L}_{total} = \mathcal{L}_{SEG} + \lambda \times (\mathcal{L}_{TP} + \mathcal{L}_{QP} + \mathcal{L}_{QM}). \tag{8}$$

Table 1: Performance on S3DIS dataset using mean-IoU metric (%). $S^i$ represents the split $i$ is used for testing. The best results are masked in **bold**. Our method consistently far exceeds SOTA.

| Method | 2-way | | | | | | 3-way | | | | | |
|---|---|---|---|---|---|---|---|---|---|---|---|---|
| | 1-shot | | | 5-shot | | | 1-shot | | | 5-shot | | |
| | $S^0$ | $S^1$ | mean | $S^0$ | $S^1$ | mean | $S^0$ | $S^1$ | mean | $S^0$ | $S^1$ | mean |
| ProtoNet [39] | 48.39 | 49.98 | 49.19 | 57.34 | 63.22 | 60.28 | 40.81 | 45.07 | 42.94 | 49.05 | 53.42 | 51.24 |
| AttMPTI [39] | 53.77 | 55.94 | 54.86 | 61.67 | 67.02 | 64.35 | 45.18 | 49.27 | 47.23 | 54.92 | 56.79 | 55.86 |
| BFG [17] | 55.60 | 55.98 | 55.79 | 63.71 | 66.62 | 65.17 | 46.18 | 48.36 | 47.27 | 55.05 | 57.80 | 56.43 |
| SCAT [36] | 54.92 | 56.74 | 55.83 | 64.24 | 69.03 | 66.63 | - | - | - | - | - | - |
| QGPNet [10] | 56.30 | 57.62 | 56.96 | 65.34 | 69.01 | 67.17 | 47.00 | 50.12 | 48.56 | 55.80 | 58.54 | 57.17 |
| 2CBR [41] | 55.89 | 61.99 | 58.94 | 63.55 | 67.51 | 65.53 | 46.51 | 53.91 | 50.21 | 55.51 | 58.07 | 56.79 |
| QGE [20] | 58.85 | 60.29 | 59.57 | 66.56 | 79.46 | 73.01 | - | - | - | - | - | - |
| QGPA [9] | 59.45 | 66.08 | 62.76 | 65.40 | 70.30 | 67.85 | 48.99 | 56.57 | 52.78 | 61.27 | 60.81 | 61.04 |
| DPA [16] | 66.08 | 74.30 | 70.19 | 71.10 | 77.03 | 74.07 | 50.67 | 59.53 | 55.10 | 64.52 | 63.34 | 63.93 |
| **Ours** | **74.04** | **77.44** | **75.74** | **76.65** | **78.22** | **77.44** | **62.77** | **70.57** | **66.67** | **67.49** | **74.68** | **71.09** |

Table 2: Performance on ScanNet dataset using mean-IoU metric (%). $S^i$ represents the split $i$ is used for testing. The best results are masked in **bold**. Our method consistently far exceeds SOTA.

| Method | 2-way | | | | | | 3-way | | | | | |
|---|---|---|---|---|---|---|---|---|---|---|---|---|
| | 1-shot | | | 5-shot | | | 1-shot | | | 5-shot | | |
| | $S^0$ | $S^1$ | mean | $S^0$ | $S^1$ | mean | $S^0$ | $S^1$ | mean | $S^0$ | $S^1$ | mean |
| ProtoNet [39] | 33.92 | 30.95 | 32.44 | 45.34 | 42.01 | 43.68 | 28.47 | 26.13 | 27.30 | 37.36 | 34.98 | 36.17 |
| AttMPTI [39] | 42.55 | 40.83 | 41.69 | 54.00 | 50.32 | 52.16 | 35.23 | 30.72 | 32.98 | 46.74 | 40.80 | 43.77 |
| BFG [17] | 42.15 | 40.52 | 41.34 | 51.23 | 49.39 | 50.31 | 34.12 | 31.98 | 33.05 | 46.25 | 41.38 | 43.82 |
| SCAT [36] | 45.24 | 45.90 | 45.57 | 55.38 | 57.11 | 56.24 | - | - | - | - | - | - |
| QGPNet [10] | 44.63 | 42.18 | 43.40 | 54.75 | 51.81 | 53.28 | 37.86 | 34.50 | 36.18 | 47.45 | 42.74 | 45.09 |
| 2CBR [41] | 50.73 | 47.66 | 49.20 | 52.35 | 47.14 | 49.75 | 47.00 | 46.36 | 46.68 | 45.06 | 39.47 | 42.27 |
| QGE [20] | 43.10 | 46.79 | 44.95 | 51.91 | 57.21 | 54.56 | - | - | - | - | - | - |
| QGPA [9] | 57.08 | 55.94 | 56.51 | 64.55 | 59.64 | 62.10 | 55.27 | 55.60 | 55.44 | 59.02 | 53.16 | 56.09 |
| DPA [16] | 62.75 | 63.04 | 62.90 | 67.19 | 64.62 | 65.91 | 61.97 | 61.72 | 61.85 | 66.13 | 64.67 | 65.40 |
| **Ours** | **75.94** | **71.92** | **73.93** | **78.42** | **78.37** | **78.40** | **70.00** | **66.61** | **68.31** | **76.73** | **68.63** | **72.68** |

## 4 Experiments

### 4.1 Experiment Settings

**Datasets.** We verify our method on S3DIS [2] and ScanNet [4]. S3DIS contains 3D-RGB point clouds collected from 272 rooms across six indoor scenes. Its label set includes 13 classes (12 semantic classes and background clutter). ScanNet collects 1,513 point clouds from 707 indoor scenes. Its label set includes 21 classes (unannotated background and 20 semantic classes). Due to the inconvenience of processing the original room which contains an excessive number of points, we follow the pre-processing strategy in [39, 9, 16] to divide each room into $1m \times 1m$ blocks and randomly sample $M = 2048$ points from each block, yielding 7,547 blocks for S3DIS and 36,350 blocks for ScanNet, respectively. Each point contains a 9D vector, including XYZ, RGB, and normalized coordinates. Following [39], semantic classes are evenly split into two non-overlapping subsets, denoted as $S^0$ and $S^1$. For meta-learning, we train our model on one fold (*e.g.*, $S^0$) and test on another fold (*e.g.*, $S^1$). Vice versa for cross-validation.

**Evaluation Metrics.** Following conventions in the 3DSeg community, we average the mean Intersection-over-Union (mIoU) across all test classes and report the final mIoU metric under the few-shot setting.

**Implementation details.** In Appendix B, we provide framework details and training details.

### 4.2 Comparison With State-of-the-Art Methods

**Results on S3DIS.** Table 1 shows the experimental results of our method compared with state-of-the-art methods on the S3IDS dataset. Our method demonstrates a substantial improvement over the baseline QGPA [9], with a large margin of 6.22%~14.59% across various splitting settings, highlighting a significant improvement for FS-3DSeg. Additionally, our method consistently exceeds all of the previous methods, particularly surpassing state-of-the-art methods DPA [16] by a margin of 1.19%~12.10% across various splitting settings, showcasing our effectiveness and superiority. The success of our approach is attributed to the integration of comprehensive LLM-generated content and reliable pseudo-query context by GCPR and PCPR, solving the problem of semantic information constraints and class information bias, as well as enhancing refinement by dual-distillation loss.

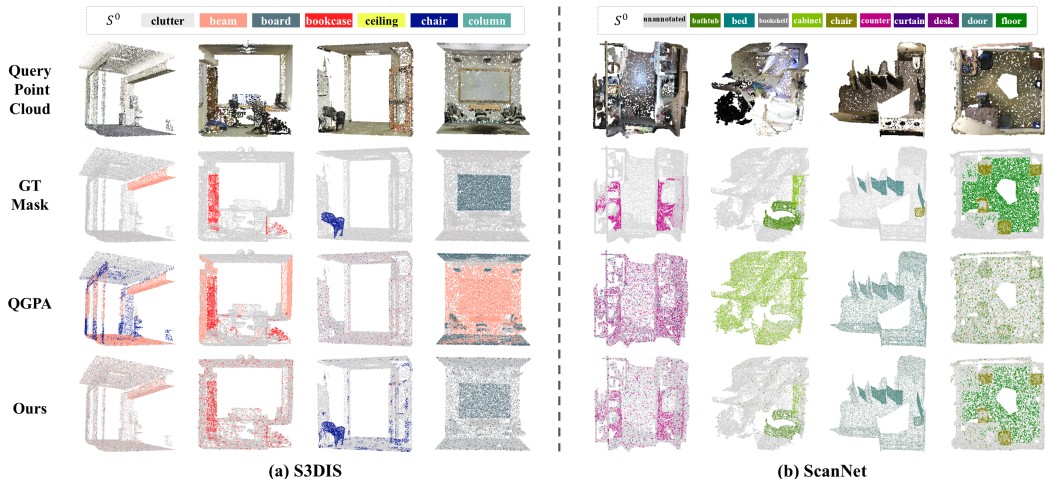

Figure 2: Qualitative results of our method on S3DIS and ScanNet in 2-way 1-shot $S^0$ segmentation task in comparison to GT mask and QGPA [9].

Table 3: Offline time cost on S3DIS and ScanNet under 2-way 1-shot setting.

| Phase | S3DIS | ScanNet |
|---|---|---|
| Description Generation: gpt-3.5-turbo | 30.23 min | 67.15 min |
| Text Feature Extraction: CLIP rn50 | 10.95 s | 17.79 s |
| Total | 30.41 min | 67.45 min |

Table 4: Analysis of online computational cost and experimental results under 2-way 1-shot setting.

| Methods | #Params | FLOPs (G) | FPS | Inference Time (ms) | S3DIS | ScanNet |
|---|---|---|---|---|---|---|
| attMPTI | 357.82K | 152.65 | 1.47 | 678.67 | 54.86 | 41.69 |
| QGPA | 2.79M | 16.30 | 38.68 | 25.85 | 62.76 | 56.51 |
| DPA | 4.85M | 15.49 | 32.35 | 30.91 | 70.19 | 62.90 |
| Ours | 4.22M | 18.96 | 20.57 | 48.61 | **75.74** | **73.93** |

**Results on ScanNet.** Table 2 presents the segmentation performance on ScanNet. Our method significantly improves the baseline by a larger margin of 11.01%~18.86%. Notably, we significantly outperform SOTA method DPA [16] by large margins of 3.96%~13.75% across various settings. Our model exhibits even greater improvement on ScanNet compared to S3DIS due to containing more classes and complex scenarios in ScanNet, which greatly challenges prototype quality. Our approach effectively integrates more comprehensive and reliable content to cope with the lack of semantic information, resulting in superior performance.

**Computational Complexity.** In our approach, generating descriptions with LLM and extracting text features through CLIP for all classes in the dataset are performed offline before the training and testing phases. Once text features are stored through offline operations, we can directly load the stored text features without regenerating by LLM online, avoiding bringing redundant computing costs during training and testing. (1) Offline cost: Table 3 shows the time cost of extracting text features for all classes in the entire dataset. We analyze that the total offline time is primarily determined by the description generation process, with feature extraction time being negligible. The time for generating descriptions from LLM varies on different datasets, depending on the number of classes. (2) Online cost: Table 4 shows that our method achieves a better balance between computational cost and experiment performance, offering superior segmentation results with reasonable computational efficiency.

**Qualitative results.** In Figure 2, we visualize the segmentation results of the 2-way 1-shot segmentation task on S3DIS and ScanNet. We compare our predictions with the GT mask and QGPA. As shown, QGPA often produces inaccurate segmentation, particularly in distinguishing background from foreground classes and among different foreground classes. In contrast, our method achieves accurate segmentation for most areas of the target scene. For instance, in the second column of S3DIS examples, QGPA mistakenly predicts parts of the beam and bookcase, whereas our method effectively distinguishes between these classes. Similar results are observed in the last column of ScanNet examples. The superiority of our method is because we incorporate LLM-generated content and pseudo-query context to obtain query-specific prototypes, resulting in better segmentation results.

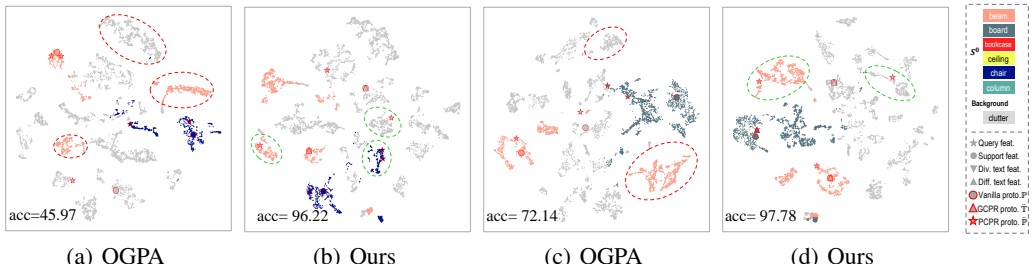

|  (a) QGPA | (b) Ours | (c) QGPA | (d) Ours |

Figure 3: Visualization of feature distribution and prototype distribution on S3DIS under 2-way 1-shot setting. 'acc' denotes the segmentation accuracy. Red/green dotted circles mark query features that are far/close to the refined prototypes, respectively. Best viewed in color.

## 4.3 Ablation Study

We perform an ablation study on the S3DIS dataset under the 2-way 1-shot setting to validate the effectiveness of our proposed GPCPR. We use QGPA [9] as the baseline to analyze the effects of the proposed components and hyper-parameters.

**Effects of Different Components.** We conduct experiments to investigate the effect of the three core components, *i.e.*, GCPR, PCPR and DD loss.

1) Effects of PCPR: Without introducing any text knowledge, PCPR improves the baseline by 8.68%, showing that PCPR extracts reliable query context, avoids noise interference, and effectively reduces class information bias.

2) Effects of GCPR: Only using diverse descriptions $D$ or differentiated descriptions $D'$, we exceed the baseline by 6.53% or 7.43%, with the latter playing a greater role because they directly improve the inter-class discrimination of the prototypes. After jointly using $D$ and $D'$, the

Table 5: Ablation study of key components on S3DIS dataset under 2-way 1-shot setting using mean-IoU metric (%).

| GCPR | | PCPR | DD loss | | | 2-way 1-shot | | |
|---|---|---|---|---|---|---|---|---|
| $D$ | $D'$ | | $\mathcal{L}_{TP}$ | $\mathcal{L}_{QP}$ | $\mathcal{L}_{QM}$ | $S^0$ | $S^1$ | **mean** |
| | | | | | | 58.96 | 63.08 | 61.02 |
| | | ✓ | | | | 65.01 | 74.39 | 69.70 |
| ✓ | | | | | | 66.06 | 69.03 | 67.55 |
| | ✓ | | | | | 66.71 | 70.18 | 68.45 |
| ✓ | ✓ | | | | | 68.57 | 74.71 | 71.64 |
| ✓ | ✓ | ✓ | | | | 68.68 | 75.73 | 72.21 |
| ✓ | ✓ | ✓ | ✓ | | | 69.36 | 75.85 | 72.61 |
| ✓ | ✓ | ✓ | | ✓ | | 71.09 | 76.09 | 73.59 |
| ✓ | ✓ | ✓ | | | ✓ | 71.07 | 76.47 | 73.77 |
| ✓ | ✓ | ✓ | ✓ | ✓ | ✓ | **74.04** | **77.44** | **75.74** |

improvement over the baseline reaches 10.62%, showing that GCPR effectively makes up for the lack of semantics of the prototype. Comparing PCPR and GCPR, the latter can achieve greater improvement because LLM provides sufficient semantic details. Finally, combining PCPR and QCPR, our method improves the baseline by 11.19%.

3) Effects of DD loss: Compared to only using PCPR and QCPR, after further applying $\mathcal{L}_{TP}$, $\mathcal{L}_{QP}$ and $\mathcal{L}_{QM}$ respectively, the model performance further improved 0.40%, 1.38% and 1.56%, with the last one being more effective due to directly optimizing pseudo masks to obtain a more reliable class-specific pseudo query context. By jointly using all distillation losses, we achieve the best performance of 75.74%.

**Visualization of features and prototypes.** As shown in Figure 4, we use the t-SNE visualization tool [30] to compare the feature distribution and the class prototype distribution to analyze the effect of the model. Here, prediction accuracy is also provided for a clearer comparison. As marked by the red dotted circles, for QGPA, the distance between prototypes and most query features is far. This indicates there exists bias between prototypes and query features due to the lack of query context in the prototypes. In contrast, our method significantly improves accuracy and pushes the prototype closer to query features, as marked by the green dotted circles. This improvement is because we effectively inject rich text knowledge and reliable class-specific query context into the refined prototypes, enhancing their representation capabilities and mitigating object variations.

**Effects of different adaptors.** As shown in Figure 4(a), without introducing any additional text information by removing the GCPR module, we evaluate the effectiveness of prototype refinement and compare our method with the previous prototype adaptors, including QGPA [9] and DPA [16]. Our method significantly outperforms QGPA and DPA. This is because although QGPA alleviates the channel-wise feature distribution gap between the prototype and query features, the prototype lacks query context information and is therefore less effective. Compared with QGPA, DPA injects query context information into the prototypes. However, query context from

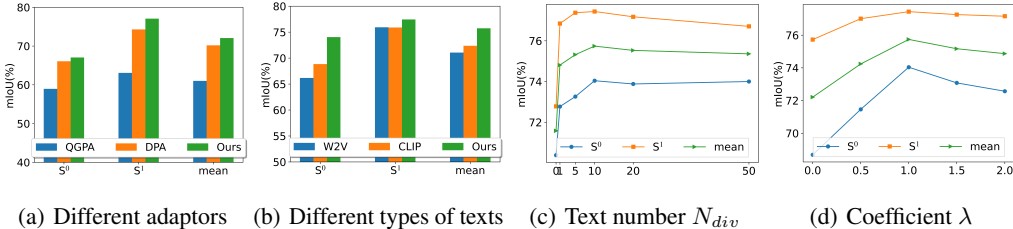

(a) Different adaptors    (b) Different types of texts    (c) Text number $N_{div}$    (d) Coefficient $\lambda$

Figure 4: Ablation study of modules and hyper-parameters on S3DIS dataset under 2-way 1-shot setting. (a) Effects of different adaptors. (b) Effects of different types of descriptions. (c) Effects of the number of LLM-generated descriptions per command. (d) Effects of coefficient $\lambda$ for DD loss.

other classes acts as noise interference, reducing the reliability of the prototype and limiting performance. Our method introduces category-specific pseudo-context information into the prototype, effectively reducing noise interference while bridging object gaps and thus achieving the highest performance.

**Effects of different types of descriptions.** As shown in Figure 4(b), we also studied the impact of different text descriptions, by replacing the proposed LLM-generated descriptions with word2vec (W2V) [19] or CLIP texts [25, 38]. As shown, word2vec exhibits limited improvement due to only providing static and restricted semantic information. CLIP texts slightly better improve the baseline, benefiting from the cross-modal alignment capability and rich semantic information learned from large-scale datasets. Our LLM-generated descriptions achieve the best performance because LLM's powerful detailed description ability can inject sufficient diverse and differentiated semantic information into the prototypes.

**Effects of the number of diverse descriptions.** Figure 4(c) illustrates the impact of the number $N_{div}$ of diverse descriptions. Compared with not using any diverse descriptions, the model performance gradually improves as $N_{div}$ increases, because general knowledge is gradually injected into the prototypes to make up for the lack of semantic information, and reaches promising results when $N_{div} = 10$. However, further increasing of $N_{div}$ does not yield higher performance as the LLM-generated descriptions begin to repeat, providing redundant semantic knowledge. In summary, we select $N_{div} = 10$ to ensure both the effectiveness and efficiency of the model.

**Effects of coefficient $\lambda$ for DD Loss.** Figure 4(d) illustrates the impact of the weight $\lambda$ of the DD loss in Eq. 8. Compared to $\lambda = 0$ (where the DD loss is deprecated), as $\lambda$ increases, the prototypes and predictions in the final stage begin to gradually optimize the corresponding items in the early stages, allowing the model to generate high-quality prototypes, resulting in significant performance improvements, and reaching the best performance when $\lambda = 1$. However, a larger $\lambda$ (*e.g.*, 2) does not yield higher performance and may even harm model performance. In summary, an appropriate value of $\lambda = 1$ yields the best results.

**Effects of different LLMs.** As shown in Table 6, using both "gpt-3.5-turbo" [3] and "gpt-4o-mini" [21] can achieve superior results compared with SOTA, with 75.74% mIoU and 73.88% mIoU on S3DIS, respectively. Specifically, Ours (gpt-3.5-turbo) performs best. Ours (gpt-4o-mini) performs slightly lower than Ours (gpt-3.5-turbo), because "gpt-4o-mini" tends to generate longer paragraphs, which potentially contain content that is not directly related to the class semantics, thus introducing noise to the class prototypes. Thus in this paper, we choose to utilize "gpt-3.5-turbo".

Table 6: Effects of different LLMs on S3DIS dataset under 2-way 1-shot setting.

| Methods | $S^0$ | $S^1$ | mean |
|---|---|---|---|
| attMPTI | 53.77 | 55.94 | 54.86 |
| QGPA | 59.45 | 66.08 | 62.76 |
| DPA | 66.08 | 74.30 | 70.19 |
| Ours (gpt-4o-mini) | 71.64 | 76.11 | 73.88 |
| Ours (gpt-3.5-turbo) | **74.04** | **77.44** | **75.74** |

## 5 Conclusion

In this paper, to solve the problem of constrained semantic information and class information bias in FS-3DSeg, we propose GPCPR that LLM-generated content and reliable query context to enhance prototype quality. GPCPR consists of two components: LLM-driven Generated Content-guided Prototype Refinement (GCPR), which enriches prototypes with diverse and differentiated text knowledge from LLMs, and Pseudo Query Context-guided Prototype Refinement (PCPR), which aggregates reliable class-specific pseudo-query context to mitigate class information bias. Additionally, a dual-distillation regularization term is proposed to facilitate knowledge transfer between early-stage and deeper entities (prototypes or predictions). Experiments show that GPCPR outperforms state-of-the-art methods by up to 12.10% and 13.75% mIoU on S3DIS and ScanNet, respectively. In Appendix C, we discuss the limitations of our work and future work.

# 6  Acknowledgement

This work was supported by the National Natural Science Foundation of China under Grant 62072027 and 62376020.

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

# A  Examples of LLM-generated Content

LLM-generated descriptions and text features for all classes are generated and stored offline. During training and testing, we directly load the stored text features, avoiding online regenerating by LLM. To more intuitively understand the diverse and differentiated descriptions generated by the LLM, we provide examples of the generated descriptions for the $N$-way $K$-shot segmentation task in Figure 5 and Figure 6, respectively.

For a single class, the descriptions generated by LLM capture rich information such as shape and geometric structure, location, detailed features, usage scenarios, and placement styles. These descriptions provide a comprehensive understanding of the class, enhancing the prototypes with detailed and varied semantic knowledge. For each class, we prompt the LLM to generate $N_{div}$ descriptions for each command. For each $N$-way $K$-shot segmentation task, we obtain $(N + 1) \times (4 \times N_{div})$ descriptions, where the number $N_{div}$ will affect the model performance, as too few or too many descriptions will lead to insufficient or redundant semantic information. Therefore, we analyze the effect of $D_{div}$ in Figure 4(c), determining that the optimal value is 10.

For class pairs, differential descriptions generated by the LLM effectively distinguish class pairs by highlighting structural differences, density differences, surface differences, and edge differences, beneficial for distinguishing similar classes. In this paper, using our proposed prompting template, the LLM tends to generate varying numbers of sentences for each class pair based on their distinguishability. We then split each sentence by conjunctions and assign each part to the relevant class. Consequently, after combining all differential sentences for each class in relation to all other classes, the number of differential descriptions $N_{diff}$ varies for each class. For each $N$-way $K$-shot segmentation task, we load descriptions based on the minimum $N_{diff}$ among the $(N + 1)$ classes. For classes with more descriptions, we randomly load $N_{diff}$ descriptions, ensuring consistent input dimensions. Thus we obtain $(N + 1) \times N_{diff}$ descriptions per task.

Intuitively, differential descriptions are more conducive to improving the discrimination of prototypes, as verified by ablation experiments in Table 5. Besides, the joint usage of both kinds of descriptions leads to the best performance in FS-3DSeg.

# B  Implementation Details

**Framework details.** We use QGPA [9] as the baseline, optimized by segmentation loss, self-reconstruction loss, and alignment loss. DGCNN [33] (with SAN) and pre-trained CLIP (rn50) [25] are used as point encoder and text encoder, respectively. The text adaptors adopt a *Linear+LeakyReLU+Dropout+Linear* architecture. In our GCPR module, the number $N_D$ of descriptions per command is set to 10. We use "gpt-3.5-turbo" GPT-3.5 engine [3] and set the temperature constant to 0.99. For diverse descriptions, the largest length of a 3D-specific diverse description is set to 40, and $N_{div}$ is set to 10. For differential descriptions, LLM generates between 1 to 20 sentences for different class pairs, causing $N_{diff}$ to vary across classes, typically ranging from 14 to 34. For all compressors based on multi-head attention, the number of heads is set to 4. In Eq. 8, the coefficient $\lambda$ for distillation loss in Eq. 8 is set to 1.

**Training details.** Our model is implemented by PyTorch and trained on single NVIDIA RTX A4000 GPU. Before meta-learning, following [39], we pre-train the point encoder on the base classes for 100 epochs. Then, during meta-training, we train our model on 40,000 episodes randomly sampled from base classes, using an Adam optimizer, with a learning rate of 0.001, and a decay step of 5000. During meta-testing, we perform model evaluation on 100 randomly sampled episodes for each feasible class combination from unseen classes. For the query set, $T$ is set to 1 for each class.

# C  Discussion

**Limitations.** Although our GPCPR achieves remarkable results, GPCPR has the following limitations. (1) Integrating LLM and CLIP into the FS-3DSeg framework will increase resource consumption and computing costs, depending on the parameter amount and execution speed of these large models. In order to alleviate this limitation, we separated some operations in GCPR into offline operations, such as prompting LLM to generate description content and using CLIP to extract text features. (2) The complexity for LLM to generate differentiated descriptions is $O(N^2)$, which may face challenges when extending to other domain datasets with massive semantic classes due to the need to exhaustively enumerate all possible class pairs.

**Future work.** The performance of the proposed method is affected by the quality of descriptions generated by LLM and the accuracy of pseudo masks. Biased class descriptions and inaccurate pseudo masks may introduce noise, negatively impacting model performance. Future work will mainly focus on improving the quality and reliability of LLM-generated content and pseudo masks. Furthermore, in this paper, we mainly address segmenting unseen categories in the same scene, but its generalization to different domains with unseen classes remains to be tested. Future research may involve extensive validation across various datasets and real-life scenarios to ensure robustness and adaptability.

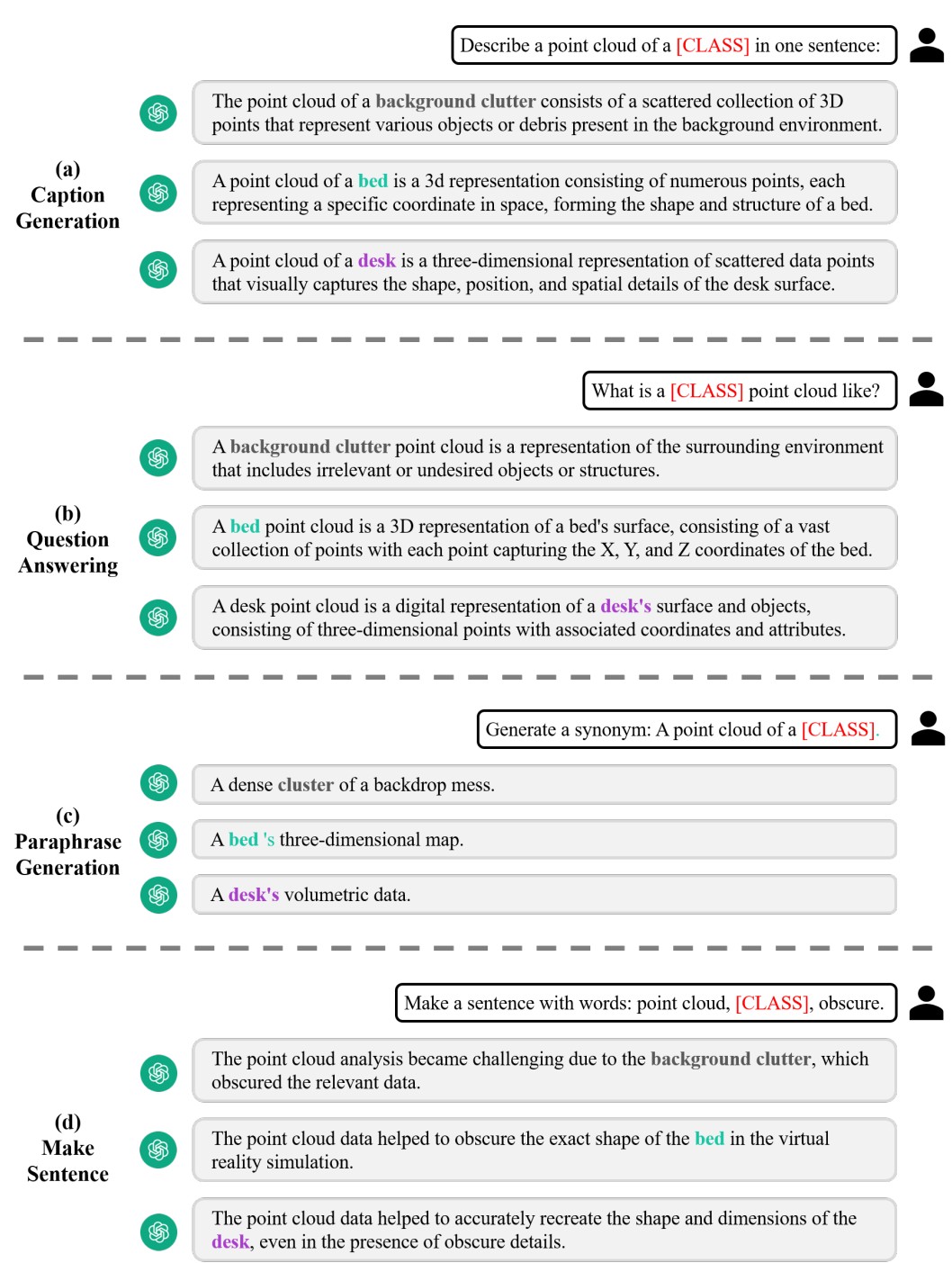

Figure 5: Example of the diverse class descriptions generated by LLM. We input four different commands to LLMs to generate diverse descriptions.

Describe the distinguishing visual or geometry differences between the point clouds of bed and desk in pairs of sentences. Generate as many captions as you can.

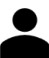

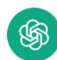

1. The point cloud of bed typically has a larger surface area with more irregular shapes and angles, while the point cloud of desk tends to have more defined and straight edges with a smaller footprint. \n

2. Beds typically have a point cloud that is lower to the ground and spread out horizontally, while desks have a point cloud that is taller and more vertical in nature. \n

3. (More captions omitted here)

Describe the distinguishing visual or geometry differences between the point clouds of bed and background clutter in pairs of sentences. Generate as many captions as you can.

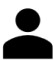

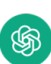

1. The bed's point cloud exhibits a more organized and structured arrangement of points, while the background clutter appears more haphazard and disorganized.

2. (More captions omitted here)

Describe the distinguishing visual or geometry differences between the point clouds of desk and background clutter in pairs of sentences. Generate as many captions as you can.

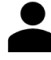

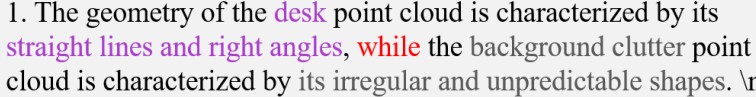

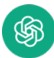

1. The geometry of the desk point cloud is characterized by its straight lines and right angles, while the background clutter point cloud is characterized by its irregular and unpredictable shapes. \n

2. The desk point cloud is composed of precise measurements and uniform spacing between points, whereas the background clutter appears fragmented and disorganized. \n

3. (More captions omitted here)

Figure 6: Example of the differentiated class descriptions generated by LLMs. For all classes, we combine any two of these classes as input to LLM to generate several differentiated sentences. Then we split each output into several sentences, which are further splited by conjunctions (*e.g.*, "while", "whereas", "compared to", "unlike", "in contrast to", "contrast", "however", and so on) and assign each part to the corresponding class.

