# OpenReview forum: "Generated and Pseudo Content guided Prototype Refinement for Few-shot Point Cloud Segmentation"
_NeurIPS.cc/2024/Conference — NeurIPS 2024 spotlight_

### Official Review · Reviewer_pd6x · 2024-07-12

**Soundness:** 2
**Presentation:** 2
**Contribution:** 1
**Rating:** 5
**Confidence:** 4

**Summary:**

This paper introduces a FS-3DSeg framework called Generated and Pseudo Content guided Prototype Refinement (GPCPR) for few-shot 3D point cloud semantic segmentation, leveraging LLM-generated content and reliable query context to enhance prototype quality. GPCPR includes two core components: Generated Content-guided Prototype Refinement (GCPR), which enriches prototypes with comprehensive semantic knowledge from large language model-generated class descriptions, and Pseudo Query Context-guided Prototype Refinement (PCPR), which mitigates class information bias by aggregating reliable class-specific pseudo-query context to create more suitable query-specific prototypes. Additionally, a dual-distillation regularization term is introduced to enable knowledge transfer between early-stage entities and their deeper counterparts, enhancing refinement. Extensive experiments demonstrate the superiority of GPCPR.

**Strengths:**

- The proposed method is technically sound.
- The experiments confirm the effectiveness of the proposed method.
- The paper is well-written

**Weaknesses:**

- The novelty is relatively limited. This paper is not the first to explore using LLM for few-shot segmentation; it has already been done in 2D few-shot segmentation [1][2]. Since LLMs inject semantic knowledge into visual features, their rich semantic knowledge should be transferable between 2D and 3D. This paper lacks specific design or optimization for 3D tasks, making the innovation of the GCPR module quite limited.
- The idea of using pseudo predictions to generate class-specific pseudo-query context in the PCPR module is quite straightforward (not novel) in few-shot segmentation. Additionally, the PCPR module simply stacks techniques from previous works. For example, QGPA directly corresponds to [3], and prototype distillation is identical to [4].
- The introduction of LLMs may significantly increase computational overhead and inference time compared to previous methods.
[1]	LLaFS: When Large Language Models Meet Few-Shot Segmentation  CVPR2024
[2]	Simple Semantic-Aided Few-Shot Learning  CVPR2024
[3]	Prototype Adaption and Projection for Few- and Zero-shot 3D Point Cloud Semantic Segmentation  TIP 2023
[4]	Dynamic Prototype Adaptation with Distillation for Few-shot Point Cloud Segmentation   3DV 2024

**Questions:**

1. The authors propose using LLM-generated content to enrich prototypes. How sensitive is the model's performance to the quality and diversity of these generated descriptions? Have the authors explored the impact of using different LLMs or prompting strategies?
2. The paper introduces a dual-distillation regularization term. How does this compare to other knowledge distillation techniques in few-shot learning? What is the specific advantage of this approach in the context of 3D point cloud segmentation?
3. The paper uses pseudo masks to extract class-specific pseudo-query context. How robust is this approach to potential errors in the initial pseudo mask generation? Have the authors explored the impact of different thresholds or strategies for generating these masks?
4. The proposed method significantly outperforms state-of-the-art approaches on both S3DIS and ScanNet datasets. Have the authors investigated why the performance gain is notably larger on ScanNet compared to S3DIS?
5. How does the computational complexity of GPCPR compare to existing methods, especially considering the integration of LLM-generated content and multiple refinement stages?

**Limitations:**

1. scalability: The complexity of generating differentiated descriptions grows quadratically with the number of classes (O(N^2)). This could pose challenges when applying the method to datasets with a large number of semantic classes.
2. Dependency on LLMs: The reliance on large language models for generating descriptions introduces an external dependency that may not always be available or consistent across different implementations or time periods.
3. Computational overhead: The paper lacks a detailed analysis of the additional computational cost introduced by the LLM content generation and multiple refinement stages. This is crucial for understanding the method's practical applicability.
4. Sensitivity to LLM output: The quality of the generated descriptions could vary depending on the specific LLM used and its training data. The paper doesn't thoroughly explore how variations in LLM output might affect the overall performance.
5. Pseudo mask reliability: The method relies on generating pseudo masks for query point clouds. However, the paper doesn't extensively analyze how errors in these pseudo masks propagate through the system and affect the final segmentation results.

---

> ### Author Rebuttal · Authors · 2024-08-07
>
> > **W1. Novelty of using LLM. Specific design.**
> 1) Novelty:
> FS-3DSeg is more complex than 2D, due to 3D data is complex. No prior work has applied LLMs to FS-3DSeg. Directly extending [1][2] to FS-3DSeg faces challenges: [1] finetunes LLMs and outputs 2D polygon coordinates, unsuitable for 3D objects; [2] lacks 3d-related descriptions. These challenges motivated us to pioneer the use of LLM to solve FS-3DSeg.
> 2) Our specific design for 3D tasks:
> ①Four 3D-oriented LLM commands for generating diverse class descriptions covering 3D properties like geometric structures and surface shape.
> ②3D-specific prompt for generating differentiated descriptions that highlight visual or geometric differences.
> ③Two Text-To-Prototype Compressors for enriching 3D prototypes with knowledge from LLM.
>
> > **W2. Novelty of PCPR.**
>
> Our PCPR is the first to use pseudo-query context to refine prototypes in FS-3DSeg. Innovations include query-to-prototype compressor, prototype distillation, and pseudo-prediction distillation, improving 3D prototype quality and pseudo mask accuracy.
>
> Difference with other methods: QGPA is used as baseline.
> [4] only distills prototypes at the last stage, while we distill prototypes at multiple progressive prototype refinement stages to ensure more stable refinement.
> Beyond that, we also propose pseudo-prediction distillation to improve pseudo masks, which have not been explored before. These designs effectively tackle semantic information bias in FS-3DSeg.
>
> > **W3. Computational Overhead.**
>
> The usage of LLMs does not increase inference time, as descriptions and text features for all classes are generated and stored offline in one phase. During training/testing, we load the stored text features, avoiding redundant computing costs for regenerating them by LLM online. Specifically, the total offline times are 30.41 minutes for S3DIS and 67.45 minutes for S3DIS and ScanNet, mainly due to the text generation process, with negligible feature extraction time (see Table 1 in the rebuttal PDF). During inference, our model better balances time costs and results, achieving superior results with moderate computational cost compared to the SOTA (see Table 2 in the rebuttal PDF).
>
> > **Q1. Impact of LLMs or prompting strategies.**
> 1) Different LLMs: Using both `gpt-3.5-turbo` and `gpt-4o-mini` can achieve superior results compared with SOTA, with 75.74% mIoU and 73.88% mIoU on S3DIS, respectively. (see Table 3 in rebuttal PDF).
> 2) Prompting Strategies: Differentiated descriptions outperform diverse descriptions by 0.9% mIoU. Combining both strategies yields best performance (see Table 3 in our paper).
> 3) Number of Descriptions: As shown in Figure 4 (c) of our paper, mIoU increases with more diverse descriptions, stabilizing around 10/command.
>
> > **Q2. Compare DD loss with other KD.**
> 1) Compared with other KD: We verify the effects of logit distillation, feature distillation, and relational knowledge distillation. Our DD loss performs better by adaptively improving the quality of pseudo masks and prototypes through multi-stage distillation (see Table 4 in rebuttal PDF).
> 2) Advantage: The complexity of 3D data leads to low-quality prototypes and inaccurate predictions. Our DD loss enables early-stage 3D prototypes or predictions to gain insights from their deeper optimal counterparts, facilitating bidirectional information exchange, and resulting in accurate segmentation results.
>
> > **Q3. Effects of pseudo mask errors.**
>
> 1) Robustness: Our method is robust to pseudo mask errors, since PCPR and DD loss enable prototypes and prediction to mutually benefit each other, adaptively improving pseudo mask quality.
> 2) Impact of thresholds: Experimental results in Table 5 in rebuttal PDF file reveal:
> - Threshold = 0: Optimal performance without filtering operation (reported in our paper).
> - Threshold <= 0.1: Comparable to Threshold=0, indicating effective integration of pseudo query context.
> - Threshold > 0.1: Performance decreases as most query features are filtered out.
>
> > **Q4. Why performance gain is larger on ScanNet?**
>
> 1) ScanNet has more classes, so the limited 3D support set is insufficient to distinguish classes. Our GCPR integrates rich semantic knowledge from LLM into 3D prototypes to improve performance.
> 2) ScanNet’s complex scenarios lead to higher class information bias. Our PCPR integrates reliable pseudo query context to refine prototypes, boosting performance.
>
> > **Q5. Computational complexity of GPCPR.**
>
> GCPR and PCPR take 12.84 ms and 11.01 ms, respectively. We achieve the best performance with moderate total computing cost (see Table 2 in rebuttal PDF).
>
> > **L1. Scalability.**
>
> In the offline stage for generating differentiated descriptions, it takes 20 minutes and 50 minutes for S3DIS and ScanNet. To address generation complexity for more classes, we provide the following alternatives:
> 1) We can divide all classes into several groups and generate differentiated text in parallel. This greatly reduces the time cost and maintains our superior performance.
> 2) We can remove the differentiated description generation process and only use diverse descriptions. By this, mIoU drops slightly from 75.74% (reported in the paper) to 73.84 % but still far exceeds SOTA (DPA 70.19%).
>
> > **L2. Dependency on LLMs.**
>
> Descriptions and text features of all classes are generated and stored in offline stage. During training/testing, we load stored text features without regeneration, thus ensuring availability and consistency across different settings or time periods. Besides, we do not rely on a specific LLM, using other LLMs (e.g., `gpt-4o-mini`) in GCPR module still far surpasses SOTA (see Table 3 in rebuttal PDF). In addition, even if no LLM is available, using alternative texts such as Word2Vec or CLIP Text still exceeds the SOTA due to the effective cooperation of our GCPR, PCPR, and DD losses (see Figure 4(b) of our paper).
>
> For L3-L5, please see our response to W3, Q1 and Q3, respectively.

---

> > ### Comment · Reviewer_pd6x · 2024-08-14
> >
> > Thank you for the response, after considering other reviews as well as the rebuttal, I decided to raise my rating to borderline accept.

---

> > > ### Author Response · Authors · 2024-08-14
> > >
> > > Dear Reviewer pd6x,
> > >
> > > Thank you very much for raising the rating to borderline accept. We sincerely appreciate your time and effort in reviewing our paper and for your constructive comments. Your insights are very valuable in helping us improve our work.
> > >
> > > Thank you once again, and we wish you all the best.

---

### Official Review · Reviewer_FCL4 · 2024-07-12

**Soundness:** 3
**Presentation:** 3
**Contribution:** 3
**Rating:** 7
**Confidence:** 4

**Summary:**

This paper analyses and addresses two issues of prototype-based methods in the Few-shot Point Cloud Segmentation task. For the constrained semantic information issue, they present the GCPR module to enrich prototypes with text knowledge via LLM and CLIP. For the class information bias issue, the proposed PCPR module mitigates this bias with reliable context aggregation. Further, the dual-distillation regularization promotes the refinement of prototypes.

**Strengths:**

This idea of enriching prototypes with knowledge from text is innovative, and the impressive performance on various datasets demonstrates its effectiveness.

The provided visualization is beneficial for explaining the proposed method.

**Weaknesses:**

As the authors mentioned in the Limitations, LLM and CLIP will bring additional computing costs. Although the proposed offline operation can help to some extent, specific metrics like model parameters and FLOPs should be provided to make a fair comparison with the previous methods. Also, I wonder whether the authors have tried to distillate knowledge from text to prototype directly like these works[1,2], which may be more efficient because there is no need to use LLM and CLIP during inference.

[1] ULIP: Learning Unified Representation of Language, Image and Point Cloud for 3D Understanding, CVPR2023

[2] OpenShape: Scaling Up 3D Shape Representation Towards Open-World Understanding, NeurIPS 2023

**Questions:**

Please refer to Weaknesses.

**Limitations:**

The authors have analyzed the limitations.

---

> ### Author Rebuttal · Authors · 2024-08-07
>
> Dear Reviewer FCL4,
>
> We thank you for taking the time to review our manuscript and offer detailed and constructive comments. We appreciate your positive reception of our work and carefully considered each of your points and would like to address your comments as follows:
>
> > **1. Computing cost and specific metrics.**
>
> Thank you for your helpful suggestion. In our approach, generating descriptions with LLM and extracting text features through CLIP for all classes in the dataset are performed offline before the training and testing phases. Once text features are stored through offline operations, we can directly load the stored text features without regenerating by LLM online, avoiding bringing redundant computing costs during training and testing. We will add the analysis of computing cost to our paper, as detailed below:
>
> **(1) Offline computation cost:** The table below shows the time cost of extracting text features for all classes in the entire dataset. We analyze that the total offline time is primarily determined by the description generation process, with feature extraction time being negligible. And the time for generating descriptions from LLM varies on different datasets, depending on the number of classes.
> | Phase | S3DIS | ScanNet |
> |:-:|:-:|:-:|
> |Description Generation: gpt-3.5-turbo|30.23 min|67.15 min|
> |Text Feature Extraction: CLIP rn50|10.95 s|17.79 s|
> |Total|30.41 min|67.45 min|
>
> **(2) Specific metrics:** The table below compares the computational costs and experimental results of our model with SOTA methods under the 2-way 1-shot setting. Our approach effectively balances computational cost and performance, achieving the highest performance with moderate computational cost.
>
> | Methods | #Params  | FLOPs (G) | FPS   | Inference Time (ms) | S3DIS | ScanNet |
> |:-:|:-:|:-:|:-:|:-:|:-:|:-:|
> | attMPTI | 357.82K  | 152.65    | 1.47  | 678.67   | 54.86 | 41.69   |
> | QGPA    | 2.79M    | 16.30     | 38.68 | 25.85    | 62.76 | 56.51   |
> | DPA     | 4.85M    | 15.49     | 32.35 | 30.91    | 70.19 | 62.90   |
> | Ours    | 4.22M    | 18.96     | 20.57 | 48.61    | **75.74** | **73.93**   |
>
> > **2. Distillate knowledge from text to prototype directly.**
>
> Thank you for your insightful comment. [1][2] follow the paradigm of 3D-CLIP contrastive pre-training and aim to distill knowledge from pre-trained CLIP to align 3D, 2D and text representations. Following them, we conduct more experiments to directly distill knowledge from text to 3D prototypes. Specifically, we remove the proposed Text-to-Prototype Compressors and Prototype Distillation Loss in the GCPR module and incorporate a contrastive loss to align 3D prototypes and text features, where the prototype is treated as a query, with text features from the same class as positive samples and those from different classes as negative samples. As you noted, this avoids the need for LLM and CLIP during inference. The experimental results are shown in the table below.
>
> | Loss Weight | 0.00  | 0.10  | 0.20  | 0.50  | 1.00  | Ours  |
> |:-:|:-:|:-:|:-:|:-:|:-:|:-:|
> | S0    | 66.17 | 68.18 | 68.43 | 65.56 | 63.45 | **74.04** |
> | S1    | 74.60 | 77.00 | 76.43 | 72.46 | 71.09 | **77.44** |
> | Mean  | 70.39 | 72.59 | 72.43 | 69.01 | 67.27 | **75.74** |
>
> We find that while 3D-text contrastive training can distill text knowledge into 3D prototypes to some extent, it is not as effective as our approach. We analyze that the success of [1][2] depends on the use of large-scale 3D datasets with a large number of classes for pre-training, whereas our FS-3DSeg task involves limited labeled data and non-overlapping training and testing classes, making such distillation methods less effective for this task. In contrast, our approach proposes two Text-to-Prototype Compressors in the GCPR module, which directly aggregate diverse and differentiated text features to enhance the quality of 3D prototypes. Additionally, the prototype distillation loss facilitates effective information transfer between prototypes at different stages, improving the prototype refinement process and leading to superior performance in the FS-3DSeg task.
>
> Thank you once again for your positive feedback, and constructive and insightful suggestions. Your feedback is crucial in helping us refine our paper.

---

> > ### Comment · Reviewer_FCL4 · 2024-08-12
> > **reply**
> >
> > The authors have addressed my concerns.
> >
> > I think this is a good paper that presents an interesting idea of enriching prototypes with knowledge from the text. Although it has some slight limitations / weaknesses, it provides a good attempt and the method is effective.

---

> > > ### Author Response · Authors · 2024-08-13
> > >
> > > Dear Reviewer FCL4,
> > >
> > > Thank you very much for your positive feedback and for taking the time to review our paper. We greatly appreciate your thoughtful comments. Your insights are very valuable in helping us improve our work.

---

### Official Review · Reviewer_ABMp · 2024-07-15

**Soundness:** 3
**Presentation:** 3
**Contribution:** 3
**Rating:** 6
**Confidence:** 5

**Summary:**

This paper targets few-shot 3D point cloud semantic segmentation and proposes GPCPR to explicitly leverage the LLM-generated content and query context to enhance the prototype quality. The component GCPR integrates diverse and differentiated class descriptions generated by LLMs to enrich prototypes. The component PCPR further aggregates reliable class-specific pseudo-query context to mitigate class information bias and generate more suitable query-specific prototypes. Furthermore, a dual-distillation regularization term is also introduced to enable knowledge transfer between early-stage entities (prototypes or pseudo predictions) and their deeper counterparts to enhance refinement. Experiments are conducted on the S3DIS and ScanNet datasets.

**Strengths:**

This paper is easy to read. This paper targets an interesting problem, few-shot 3D point cloud semantic segmentation. The proposed method is also interesting.

**Weaknesses:**

Figure 1 should explain what the symbols mean, such as P, T, S, and dot.

The CVPR 2024 paper [1] found that there are some issues in the experimental setting of few-shot 3D point cloud segmentation and corrected the setting. However, this paper is still evaluated under the old setting and does not compare with [1], which makes the results less convincing. Thus, I believe that the experiments must be carried out under the new setting in the final version.

In Table 3, the PCPR module seems to only yield a marginal improvement. Please provide some explanations.

Considering that this paper uses LLMs, which classes does the model perform best and which classes does the model perform worst? Is there any connection between this finding and LLMs?

**Questions:**

What is the performance of the proposed method under the setting corrected by [1]?

**Limitations:**

The experiments are carried out in the old, ill-posed setting.

---

> ### Author Rebuttal · Authors · 2024-08-07
>
> Dear Reviewer ABMp,
>
> We would like to express our gratitude for taking the time to review our manuscript and providing detailed and constructive feedback. We appreciate your positive reception of our work and carefully considered each of your points and would like to address your comments as follows:
>
> > **1. Explain symbols in Figure 1.**
>
> Thank you for your valuable suggestion. We will add legends to Figure 1 to clearly explain the symbols. Specifically, $\mathbb{P}$ denotes the original 3D prototype set. $\dot{\mathbb{T}}$, $\ddot{\mathbb{T}}$, $\dot{\mathbb{P}}$ and $\ddot{\mathbb{P}}$ represent prototype sets at different refinement stages. Among them, $\dot{\mathbb{T}}$ and $\ddot{\mathbb{T}}$ represent prototype sets refined by diverse and differentiated descriptions, $\dot{\mathbb{P}}$ and $\ddot{\mathbb{P}}$ indicate prototype sets refined by QGPA module and Query-to-Prototype Compressor, respectively. $S$ denotes cosine similarity, and $\odot$ represents matrix dot product.
>
> > **2. Experiments under the new setting of [1].**
>
> Thank you for your valuable comments. Based on the new experimental settings corrected by CVPR 2024 paper [1], we conduct more experiments under the 2-way 1-shot setting on S3DIS dataset. The results are presented in the table below. Our method achieves slightly higher performance than COSeg but far exceeds attMPTI and QGPA, demonstrating the proposed GCPR module, PCPR module and DD-loss can collaborate to improve the prototype quality by integrating comprehensive text descriptions and reliable pseudo query context. We will include these experimental results and analysis in the final version of our paper to provide a more convincing comparison with SOTA under the new setting.
>
> Methods | $S^0$ |$S^1$ | mean
> :--------| :---------:|--------:|--------:
> attMPTI | 31.09 | 29.62 | 30.36|
> QGPA | 25.52 | 26.26 | 25.89
> COSeg | 37.44 | 36.45 | 36.95
> Ours | 37.96 | 37.38 | 37.67
>
> > **3. In Table 3, the PCPR module seems to only yield a marginal improvement. Please provide some explanations.**
>
> The impact of the PCPR module depends on the scale of the dataset, the number of classes, and the complexity of scenes.
>
> (1) In Table 3, on the **S3DIS** dataset, without GCPR (Row 2 vs. Row 1), PCPR significantly improves the baseline by 8.68% mIoU due to its effective handling of class information bias between support and query sets. With GCPR (Row 6 vs. Row 5), PCPR only further yields a 0.64% improvement because GCPR already addresses the semantic information constraints of support set in S3DIS dataset with less classes and simpler scenes.
>
> (2) On the **ScanNet** dataset, removing PCPR causes a 5.33% drop in mIoU under 2-way 1-shot setting (68.60% without PCPR vs. 73.93% with PCPR), highlighting the importance of PCPR in reducing class information bias in complex scenes with more classes.
>
> In summary, PCPR is more impactful on large-scale dataset with greater complexity and more classes, such as ScanNet. We will explain this more clearly in the paper.
>
> > **4. Which classes perform best/worst? Is there any connection between this finding and LLMs?**
>
> Thank you for your insightful question.
>
> (1) The **class performance** of our model is shown as below:
>
> * S3DIS dataset: Best on *"chair"* (or *"sofa"*) and worst on *"ceiling"* (or *"wall"*) under $S^0$ (or $S^1$) setting.
>
> * ScanNet dataset: Best on *" curtain "* (or *"sofa"*) and worst on *"floor"* (or *"picture"*) under $S^0$ (or $S^1$) setting.
>
> For example, under the 2-way 1-shot setting, the class performance are as follows:
>
> | | S3DIS  |  | ScanNet|  |
> |:-:|:-:|:-:|:-:|:-:|
> | | S0 | S1 | S0 | S1 |
> | Best IoU (%) | chair: 87.18 | sofa: 89.40 | curtain: 86.11 | sofa: 82.09 |
> | Worst IoU (%) | ceiling: 61.08| wall: 67.40| floor: 64.08 | picture: 51.89 |
>
> We observe that our model excels in classes with complex geometric structures, unique shapes, and rich colors, but performs not as well in classes with flat geometric structures. Notably, compared to the baseline—which performs best on *"chair"* (or *"sofa"*) and worst on *"beam"* (or *"table"*) on S3DIS dataset under the $S^0$ (or $S^1$) setting—our model consistently shows improvements in both per-class IoU and mIoU. This is particularly evident in classes with complex structures, such as *"beam"* and *"table"*, where performance increases from 32.06% to 63.50% and from 53.22% to 72.39%, respectively.
>
> (2) **Connections with LLM**: The superior performance of our model on complex structures can be attributed to the introduction of LLM, which can generate more diverse and distinguishable visual or geometric descriptions for classes with complex structures, resulting in higher performance. Conversely, for classes with flat structures, LLM produces less rich descriptions that struggle to capture subtle differences, resulting in not as good performance.
>
> Once again, we thank you for your positive feedback, constructive criticism, and thoughtful suggestions. Your feedback plays an integral role in refining our work.

---

### Author Rebuttal · Authors · 2024-08-07

To PC, AC, and all Reviewers:

We sincerely appreciate the time and effort the PC, AC, and all reviewers have dedicated to reviewing our work. We are grateful for the detailed and thoughtful feedback on our submission, particularly the positive comments and insights. Below, we summarize the strengths recognized by the reviewers:

1. **Innovation**: The proposed method is interesting, innovative and technically sound. (Reviewer ABMp , Reviewer FCL4, Reviewer pd6x)
2. **Effectiveness**: The impressive performance on various datasets demonstrates the effectiveness of the proposed method. (Reviewer FCL4, Reviewer pd6x)
3. **Readable**: The paper is well-written and easy to read. (Reviewer pd6x, Reviewer ABMp)
4. **Beneficial visualizations**: The paper includes beneficial visualizations for explaining the proposed method. (Reviewer FCL4)

Thank you once again for your time and effort in reviewing this manuscript. For the questions and other concerns raised by the reviewer, we respond to each in detail and outlined our plans for improvement. If any questions are not answered or our response is unclear, we would appreciate the opportunity to communicate further with our reviewer.

---

### Decision · Program_Chairs · 2024-09-25

**Decision:**

Accept (spotlight)

**Comment:**

Generally the reviewers agree on the contribution, e.g. from one of the reviews: This idea of enriching prototypes with knowledge from text is innovative, and the impressive performance on various datasets demonstrates its effectiveness.  The provided visualization is beneficial for explaining the proposed method.  Also the reviewers agree on the potential downside of this approach relying on LLMs and the concomitant issues and compute required.  Overall accept.